# TGF-β1 activates neutrophil signaling and gene expression but not migration

**Lauren E. Hein**[1,2]**, Shuvasree SenGupta**[3]**, Gaurie Gunasekaran**[3,4]**, Craig N. Johnson**[5]**, Carole A. Parent**[1,2,3,5,6]*

**1** Cancer Biology Graduate Program, University of Michigan Medical School, Ann Arbor, MI, United States of America, **2** Rogel Cancer Center, University of Michigan, Ann Arbor, MI, United States of America, **3** Life Sciences Institute, University of Michigan, Ann Arbor, MI, United States of America, **4** LS&A Program in Biology, University of Michigan, Ann Arbor, MI, United States of America, **5** Department of Cell and Developmental Biology, University of Michigan Medical School, Ann Arbor, MI, United States of America, **6** Department of Pharmacology, University of Michigan Medical School, Ann Arbor, MI, United States of America

* parentc@umich.edu

**Data Availability Statement:** The raw RNAseq FASTQ files and the counts matrix are available in GEO accession GSE230490. All other relevant data are within the paper and its Supporting Information files.

## Abstract

Tumor-associated neutrophils are found in many types of cancer and are often reported to contribute to negative outcomes. The presence of transforming growth factor-beta (TGF-β) in the tumor microenvironment reportedly contributes to the skewing of neutrophils to a more pro-tumor phenotype. The effects of TGF-β on neutrophil signaling and migration are, however, unclear. We sought to characterize TGF-β signaling in both primary human neutrophils and the neutrophil-like cell line HL-60 and determine whether it directly induces neutrophil migration. We found that TGF-β1 does not induce neutrophil chemotaxis in transwell or underagarose migration assays. TGF-β1 does activate canonical signaling through SMAD3 and noncanonical signaling through ERK1/2 in neutrophils in a time- and dose-dependent manner. Additionally, TGF-β1 present in the tumor-conditioned media (TCM) of invasive breast cancer cells results in SMAD3 activation. We discovered that TCM induces neutrophils to secrete leukotriene $B_4$ ($LTB_4$), which is a lipid mediator important for amplifying the range of neutrophil recruitment. However, TGF-β1 alone does not induce secretion of $LTB_4$. RNA-sequencing revealed that TGF-β1 and TCM alter gene expression in HL-60 cells, including the mRNA levels of the pro-tumor oncostatin M (*OSM*) and vascular endothelial growth factor A (*VEGFA*). These new insights into the role and impact of TGF-β1 on neutrophil signaling, migration, and gene expression have significant implications in the understanding of the changes in neutrophils that occur in the tumor microenvironment.

## Introduction

Neutrophils are an essential component of the innate immune system, helping to protect the body by responding to sites of infection or injury. They do so by executing a number of functions, including phagocytosis of pathogens or cell debris, release of cytotoxic enzymes or reactive oxygen species (ROS), and release of webs of DNA known as neutrophil extracellular traps

**Funding:** This work was supported by funding from the University of Michigan School of Medicine to CAP, NIH (National Institutes of Health) grant R01 AI152517 to CAP, and the Rackham Predoctoral Fellowship to LEH. The funders had no role in study design, data collection and analysis, decision to publish, or preparation of the manuscript.

**Competing interests:** The authors have declared that no competing interests exist.

(NETs) [1, 2]. To reach inflamed or infected sites, neutrophils sense and respond to local chemoattractants released at these sites in a process known as chemotaxis [3]. In response to these chemoattractants, activated neutrophils produce and respond to the secondary chemoattractant leukotriene B$_4$ (LTB$_4$) to recruit distant neutrophils in a process known as signal relay [4–6].

Neutrophils are also recruited to many types of tumors, including breast cancer [7–9]. Interestingly, tumor-associated neutrophils (TANs) have been detected more frequently in aggressive triple negative breast cancers (TNBCs) compared to their hormone receptor positive (HR+) counterparts [10, 11]. However, in the context of cancer, many of the beneficial functions of neutrophils stimulate tumor progression [12], including immunosuppression, remodeling of the extracellular matrix, promotion of angiogenesis, and release of NETs [13–19]. The pro-tumor phenotype of neutrophils is at least partially mediated through TGF-β [20–23], a cytokine frequently present in tumors [24]. It has been shown that neutrophils isolated from tumors in mice are less cytotoxic and have a more immunosuppressive transcriptome compared to neutrophils isolated from tumors that are treated with TGF-β blockade [20, 22]. However, these *in vivo* studies are performed in the presence of a TGF-β inhibitor applied systemically, making it difficult to determine the signaling kinetics and the direct impact of TGF-β on neutrophil responses.

We previously reported that tumor conditioned media (TCM) harvested from highly aggressive triple negative breast cancer (TNBC) cell lines gives rise to robust neutrophil recruitment activity, compared to TCM harvested from poorly aggressive HR+ breast cancer cell lines [25]. By blocking TGF-β signaling in combination with inhibition of the C-X-C motif chemokine receptor 2 (CXCR2), we reported a significant reduction in neutrophil migration toward TCM from the aggressive breast cancer cell line MCF10CA1a (M4) [25]. Numerous studies have shown that CXCR2 ligands, such as the chemokine CXCL1, are important for neutrophil migration to tumor sites [7], and others have shown both positive and negative effects of TGF-β blockade on neutrophil migration to tumors [20, 26]. However, we found no significant reduction in migration when CXCR2 or TGF-β signaling were blocked independently [25]. We detected an abundant presence of TGF-β1 in M4 TCM, which induces robust neutrophil chemotaxis [25]. However, whether TGF-β induces neutrophil chemotaxis on its own is unclear. Some studies have reported that TGF-β1 has no effect on neutrophil chemotaxis [27], while others have shown that TGF-β1 can induce a chemotactic response in neutrophils at very low concentrations [28, 29].

In the present study, using primary neutrophils from healthy human individuals and the neutrophil-like HL-60 cell line, we evaluated neutrophil migration toward TGF-β1. Because LTB$_4$ release is an important part of neutrophil migration, we also investigated LTB$_4$ secretion from neutrophils in response to TCM and TGF-β1 and assessed the role of LTB$_4$ secretion in regulating neutrophil migration towards tumor-secreted factors. In addition, we characterized the signaling kinetics of TGF-β1 in neutrophils and evaluated the transcriptional effects of TGF-β1. Overall, we found that TGF-β1 activates neutrophils and gene transcription, but it does not induce migration or release of LTB$_4$.

## Materials and methods

### Ethics statement

Human neutrophils were isolated from blood obtained from anonymous healthy human donors form the Platelet Pharmacology and Physiology Core at the University of Michigan. The blood was attained through an institutional review board-approved (IRB#HUM00107120) protocol specifically approved to provide de-identified blood for research purposes. We

therefore did not have access to the HIPAA information. All subjects were consented, agreed to provide their blood for research purposes and were financially compensated.

## Cell cultures

The human promyelocytic leukemia HL-60 cell line (ATCC #CCL-240) was maintained in IMDM media (Gibco) with 10% heat-inactivated (HI) fetal bovine serum (FBS) (GeminiBio #100–106) and split to 0.2E6 cells/ml in fresh media every other day. Cells were used for no more than 10 passages. HL-60 cells were differentiated into neutrophil like cells, referred to as dHL-60 cells, over the course of 4 days. The cells were cultured at a starting concentration of 0.4E6 cells/ml and treated on day 0 and day 2 with fresh IMDM differentiation media supplemented with 2% HI FBS, 1.3% DMSO, 10 μg/ml insulin, 5.5 μg/ml transferrin, and 6.7 ng/ml sodium selenite [30, 31]. Tubes coated with 1% BSA were used for experiments using dHL-60 cells.

The MCF10CA1a cell line (M4), which is an invasive derivative of the human MCF10A cell line and a representative of triple negative breast cancer, was obtained from Karmanos Research Institute. M4 cells are maintained in DMEM/F12 media with 5% HI HS (Gibco #26050088).

## Isolation of human neutrophils (polymorphonuclear cells—PMNs)

Neutrophils were obtained and isolated as previously described [25, 32]. Blood was obtained through the Platelet Pharmacology and Physiology Core at the University of Michigan. The Core has a blanket IRB that allows them to collect blood from male and female subjects who are healthy and aged 19–65 years. We receive deidentified blood samples for neutrophil isolation.

Briefly, the majority of red blood cells were removed using dextran sedimentation (Sigma-Aldrich #31392). Then, the remaining cells were separated using Histopaque-1077 density centrifugation (Sigma-Aldrich #H8889). The neutrophil pellet was then cleared of remaining red blood cells using ACK lysis buffer (Lonza #10-548E). For subsequent experiments, neutrophils were resuspended in mHBSS ((20 mM HEPES, pH 7.4, 150 mM NaCl, 4 mM KCl, 1.2 mM MgCl2, and 1 mg/ml glucose).

Tubes coated with 0.5–1% BSA were used any time PMNs were used in experiments. A quality control assay is performed each time that PMNs are isolated, assessing at cell polarization in response to several doses of the bacterial peptide N-formylmethionine-leucyl-phenylalanine (fMLF). Cells that have a low basal activity to the vehicle control and respond in a dose-dependent manner to fMLF are used in subsequent experiments.

## Harvesting conditioned media from cell culture

Generation of tumor-conditioned media (TCM) was done as previously described [25]. Briefly, M4 cells were seeded at a density of 0.15E6 cells/ml in T150 culture dishes in 32 ml of full media (DMEM/F12 media with 5% HI HS) and incubated for 24 hours. The culture medium was removed, and the cells were washed twice with DPBS. 32 ml of serum-free DMEM/F12 was added to the cells for additional 48 hours. The media was then harvested and filtered through a 0.22 μm membrane filter. Aliquots were frozen at -30˚C.

## Pharmacological inhibition

Cells were pretreated with the pharmacological inhibitors for 30 minutes while rotating at 37˚C. Treatments were performed at 4E6 cells/ml in either mHBSS (PMNs) or IMDM (dHL-

60 cells). Following pretreatment, inhibitors were present throughout the entirety of the respective assays. SB431542 was the TβR1 (ALK4/5/7) inhibitor used at concentrations between 0.01 and 10 μM [33]. AZD5069 was the CXCR2 antagonist used at a concentration of 1 μM [34]. The FLAP inhibitor used was MK886 at a concentration of 100 nM [35].

## Transwell assay

Transwell assays were performed as previously described to assess PMN migration [25]. Briefly, transwell inserts with 3-μm pores and wells of a 24-well plate were coated with 2% BSA for 1 hr at 37˚C and washed twice with DPBS. PMNs were pretreated with various inhibitors or vehicle controls for 30 minutes while rotating at 37˚C. 0.4E6 PMNs were then plated onto the transwell membranes in 100 μl and allowed to migrate toward TCM for 2 hr at 37˚C. Any inhibitors used to pretreat the neutrophils were also present in the TCM for the entirety of the assay. After 2 hours, the number of cells that had migrated to the bottom chamber were counted using a hemocytometer, and the percentage of cells migrated was calculated.

## Underagarose assay

Underagarose assays were used to assess neutrophil migration and performed as previously described [36]. Briefly, 400 μl of 0.5% SeaKem ME agarose (Lonza #50010) dissolved in 50% mHBSS/50% DPBS was added to each well of an 8-well chamber slide (Cellvis #C8-1.5H-N) that had been coated with 1% BSA. The agarose solidified for 15 minutes at room temperature, followed by an additional 30 minutes at 4˚C. Two holes 1 mm in diameter and 2 mm apart were punched into the agarose of each well using a 3D-printed metal hole punch tool. After removing the agarose plugs, 50,000 cells stained with Hoechst (Invitrogen #H21492) were added to one of the punched holes, and chemoattractant was added to the other well (7 μl each). The slide was incubated at 37˚C for 2 hours, and endpoint images were taken using a 5x objective on a Zeiss Axiovert microscope. Endpoint images were quantified in ImageJ by counting the number of cells that left the well on the side closest to the chemoattractant.

## LTB4 ELISA assay

The LTB$_4$ ELISA Assay was performed per the manufacturer's instructions (Cayman Chemical #520111 or R&D Systems #KGE006B). Supernatants from PMNs were generated as previously described [4]. Briefly, primary neutrophils resuspended at 4E6 cells/ml were placed on ice for 30 minutes to bring them to a basal state. Cells were then spun at 400 xg for 5 minutes at room temperature and primed by resuspending them in warm phenol-red free RPMI (Gibco #11835–030) supplemented with 20 mM HEPES and 10 ng/ml GM-CSF for 1 hour in rotation at 37˚C. Following priming, cells were quickly spun (6000 xg for 30 sec) and stimulated with various stimuli: 500 pg/ml TGF-β1, 1 μg/ml CXCL1, 20 nM IL-8, and 20 nM fMLF diluted in RPMI + HEPES or 100% TCM and DMEM/F12 media control for 15 minutes. After stimulation, cells were quickly spun as before and the supernatant was collected and frozen at -80˚C. 20 nM fMLF was the positive control for this assay, and only experiments where LTB$_4$ secretion in response to fMLF was greater than 50 pg/ml were included.

## Western blotting analysis

Western blotting was used to assess ERK and SMAD3 phosphorylation. To prepare samples, PMNs or dHL-60 cells at 4E6 cells/ml were pretreated with 2 mM Pefabloc, a serine protease inhibitor (Sigma Aldrich), for 15 minutes at 37˚C while rotating. For pharmacological inhibitors, Pefabloc was added to cells during the last 15 minutes of the total 30 min. Cells were then

stimulated with either 2 ng/ml TGF-β or M4 TCM. At the designated time points, cells were lysed in RIPA (Fisher Scientific #50-103-5430) with 1x Halt™ protease and phosphatase inhibitor cocktail (Thermo Scientific #78442) at a density of 14 ul lysis buffer per 1E6 cells. Before loading samples onto gels, samples were mixed with 6x reducing Laemmli (Fisher Scientific #AAJ61337AD) and boiled at 95˚C for 10 minutes. Equal volumes of each sample, therefore representing equal number of cells, were loaded and run on either 4–12% tris-glycine polyacrylamide gels (Invitrogen #XP04125BOX) or 10% SDS-polyacrylamide gels using tris-glycine SDS running buffer and then transferred onto either nitrocellulose or PVDF membranes. Membranes were blocked with 1x fish gelatin (Biotium #22010) for 1 hour at room temperature and probed with primary antibodies: anti-phospho p44/42 MAPK (Erk1/2) (Thr202/Tyr204) (dilution 1:1000, CST #9101), anti-phospho-SMAD3 (S423 + S425) (dilution 1:1000, Boster Bio #P00059-1), anti-GAPDH-HRP (H12) (dilution 1:3000, SCBT #sc-166574-HRP), and anti-alpha tubulin-HRP (dilution 1:12000, Proteintech # HRP-66031). Bands were visualized using HRP-conjugated secondary antibodies (dilution 1:10,000, Jackson ImmunoResearch), SuperSignal™ West Pico PLUS Chemiluminescent Substrate, and an Azure C600 (a digital imaging system). Band intensities were quantified using the Gel Analyzer tool in ImageJ. For phospho-SMAD3 quantification, the lower of the two bands was quantified (52 kD), as did Jeon *et al* [37]. The upper band is likely phospho-SMAD2 because its observed molecular weight is 58 kD [38] and because it is phosphorylated at the SSXS motif at the C-terminus, similar to SMAD3 [39–41].

## RT-qPCR

To evaluate gene expression, RNA was isolated from dHL-60 cells using the RNeasy Mini Kit (Qiagen #74104). RNA was converted into cDNA using the High-Capacity cDNA Reverse Transcription Kit (Applied Biosystems #43-688-14) using 2 μg RNA per 20 μl reaction. The thermal cycler program used was 10 minutes at 25˚C, 120 minutes at 37˚C, and 5 minutes at 85˚C, followed by a hold at 4˚C. cDNA was stored at -30˚C. qPCR reactions were 10 μl, with PowerUp™ SYBR™ Green Master Mix, 0.3 μM of each forward and reverse primers (Table 1), 150 ng cDNA, and nuclease-free water. qPCR was run on a QuantStudio™ 5 using the following program: an initial hold at 95˚C for 10 minutes; 40 cycles of 95˚C for 15 seconds and 55˚C for 1 minute; and a melt curve analysis using temperatures from 60˚C to 95˚C. Primers were designed using IDT's PrimerQuest™ Tool. For analysis, change in gene expression was calculated using 2^-ddCt, using RACK1 [42] or 18S as the housekeeping gene.

## Bulk RNA-sequencing

RNA was extracted from dHL-60 cells using the RNeasy Mini Kit (Qiagen #74104) with the RNase-Free DNase Set (Qiagen #79254) to remove any residual DNA. Quality control (RNA-integrity number (RIN)), library prep (poly-A enrichment), and next-generation sequencing (NovaSeq 6000 S4) was carried out in the Advanced Genomics Core at the University of Michigan.

**Table 1. qPCR primers used in this study.**

| Gene | Forward primer (5' to 3') | Reverse primer (5' to 3') |
| --- | --- | --- |
| RACK1 | CGAAGGCAAACACCTTTACAC | GATCTTTCCCTCTAAATCCCAGAT |
| 18S | GCTTAATTTGACTCAACACGGGA | AGCTATCAATCTGTCAATCCTGTC |
| PAI1 | GAGGAGATCATCATGGACAGAC | GGTCAGGGTTCCATCACTT |
| KLF10 | CAGGATGTGGCAAGACATACTT | ACCTCCTTTCACAACCTTTCC |
| SMAD7 | CCTTCCTCCGCTGAAACA | CACCAGTGTGACCGATCC |

## Bioinformatics

150bp paired-end FASTQ files from the University of Michigan's Advanced Genomics Core were first trimmed using default settings with the Trim_Galore v0.6.6 wrapper around Cutadapt v3.7 to remove any remaining adapter. Reads were then mapped using STAR v2.7.10a [43] to the GRCh38.v40 reference supplied by gencodegenes.org; all samples ranged from 89–92% uniquely mapped.

Features were counted using the featureCounts function from the Rsubread Bioconductor package [44]. The GTF file supplied with the GRCh38.v40 reference was used to define features. Normalized counts per million were generated using edgeR [45] and the trimmed mean of M-values (TMM) method was used to create final Log2 transformed expression values.

Linear models were fit to each feature using the limma package of Bioconductor with the precision weights "voom" technique used to model the mean-variance relationship [46]. RNA extraction date was used as a pairing variable for a final cell means model of ~0 + group + pair. Comparisons were then calculated between all groups and time 0 as well as between treatment groups and their respective media control. P-values for multiple comparisons were adjusted using the Benjamini and Hochberg technique. Adjusted P-values less than 0.1 and a fold change greater than 1.5 was used to define significant gene lists.

Reactome pathways [47] of Table 2 were selected for downstream investigation based on prior expectations and for validation of the gene lists. Fisher's Exact tests were used to test the null hypothesis that the gene list had as many pathway genes as we would expect by chance. The gene universe consisted of all features in the GTF with mappings to Entrez Gene IDs. P-values for these 10 reactome pathways were adjusted using the Bonferroni method.

## Statistical analysis

GraphPad Prism was used to generate plots and to conduct statistical analysis. Specific tests used are mentioned in the figure legends. Samples with N = 2 were excluded from one-way ANOVA analysis.

## Results

### TGF-β1 treatment does not induce neutrophil chemotaxis

To test if TGF-β1 induces neutrophil chemotaxis, we performed both transwell and underagarose migration assays in response to recombinant TGF-β1. In the transwell assay, migration of human polymorphonuclear neutrophils (PMNs) was assessed in response to TGF-β1 at doses

**Table 2. Overrepresentation analysis of TGF-β1 vs IMDM differentially expressed genes within reactome pathways.**

| Reactome pathway | Reactome ID | Pathway genes (significant) | Bonferroni Adjusted P-Value (Fishers Exact Test) |
|---|---|---|---|
| Class A1 (rhodopsin-like receptors) | R-HSA-373076.8 | 156 (8) | 0.000003636 |
| Signaling by interleukins | R-HSA-449147.11 | 333 (12) | 0.000004578 |
| Signaling by TGF-β family members | R-HSA-9006936.5 | 84 (4) | 0.029199 |
| $G_q$ alpha signaling events | R-HSA-416476.6 | 102 (4) | 0.05991 |
| $G_i$ alpha signaling events | R-HSA-418594.7 | 145 (3) | 1 |
| Chemokine receptors bind chemokines | R-HSA-380108.4 | 24 (1) | 1 |
| Arachidonic acid metabolism | R-HSA-2142753.5 | 27 (1) | 1 |
| Neutrophil degranulation | R-HSA-6798695.2 | 428 (2) | 1 |
| MAPK family signaling cascades | R-HSA-5683057.3 | 234 (1) | 1 |
| Exocytosis of secretory granule membrane proteins | R-HSA-6798743.1 | 67 (0) | 1 |

ranging from 0.1–10 ng/ml or TCM derived from the M4 cell line, which has robust neutrophil recruiting activity [25]. We found that while about 30% of neutrophils migrated toward the positive control M4 TCM, no significant migration was noted in response to any of the TGF-β1 doses tested (Fig 1A). In the underagarose assay, we assessed the migration of differentiated HL-60 (dHL-60) cells in response to TGF-β1 or N-formylmethionine-leucyl-phenylalanine (fMLF), a potent neutrophil chemoattractant. While we observed that a significant number of dHL-60 cells migrated towards fMLF (Fig 1Bii and 1C) compared to buffer control (Fig 1Bi and 1C), no significant difference was detected in the number of cells migrating towards any of the TGF-β1 concentrations tested (Fig 1Biii-vi and 1C). Together, these findings obtained using two different migration assay systems with both primary and cell line-derived neutrophils show that TGF-β1 does not induce neutrophil chemotaxis.

## TGF-β1 activates canonical and noncanonical signaling pathways in neutrophils

Canonical TGF-β signaling occurs through the phosphorylation of the C-terminus of SMAD2/3 (pSMAD3C). A complex between pSMAD3C and SMAD4 subsequently enters the nucleus to regulate gene transcription [48]. To test if neutrophils respond to TGF-β1, we treated dHL-60 cells with increasing concentrations of TGF-β1 and measured the extent of SMAD3 phosphorylation. We found a dose-dependent increase in pSMAD3C after 30 min TGF-β1 treatment (Fig 2A and 2B). In a 90-min time course, dHL-60 cells treated with TGF-β1 showed low basal pSMAD3C that rapidly increased within 10 min and was sustained for about 60 min (Fig 2C and 2D). Similar kinetics were observed in PMNs (Fig 2E and 2F). To confirm that the response was dependent on TGF-β1, we pretreated dHL-60 cells with the TGF-βRI inhibitor SB431542 [33]. We found that concentrations of SB431542 ranging from 0.01–1 μM decreased TGF-β1-induced pSMAD3C in a dose-dependent manner, confirming that the signaling is TGF-βRI specific (Fig 2G and 2H). Taken together, these findings show that TGF-β1 induces pSMAD3C in PMNs and dHL-60 cells.

In addition to the SMAD-dependent canonical pathway, TGF-β signals through noncanonical pathways in various types of epithelial cells [49]. One of the noncanonical signaling pathways activates ERK1/2 [49]. Because signaling through ERK1/2 is involved in the regulation of neutrophil migration [50], we investigated if TGF-β1 signaling activates ERK1/2 in neutrophils by monitoring its phosphorylation status over time. When PMNs were treated with 2 ng/ml TGF-β1, we found a ~2-fold increase in pERK1/2 levels after 10–20 min, followed by a gradual decrease over the ensuing 90 min (Fig 3A and 3B). In dHL-60 cells, we detected a high basal pERK1/2 that gradually went down over time in response to TGF-β1 treatment (Fig 3C and 3D). The high basal pERK1/2 in dHL-60 cells is likely due to a mutation in the gene encoding neuroblastoma RAS viral oncogene homolog (NRAS) [51]. Together, these results establish that TGF-β1 noncanonically induces moderate activation of ERK1/2 in PMNs.

## TGF-β1 present in TCM harvested from invasive breast cancer cells activates SMAD3 signaling in neutrophils

To determine the relevance of TGF-β signaling in the context of the tumor-secreted factors of invasive breast cancer cell lines [25], we set out to assess SMAD3 and ERK1/2 activation in neutrophils treated with M4 TCM over time. As expected, we measured an increase in the levels of pSMAD3C in TCM-treated dHL-60 cells and PMNs (Fig 4A–4D), although the kinetics were slower in both cell types compared to TGF-β1 treatment alone (compare with Fig 2E–2H). To determine if the M4 TCM-dependent phosphorylation of SMAD3 is due to the TGF-β1 present in the TCM, we assessed the effects of SB431542 on M4 TCM-dependent

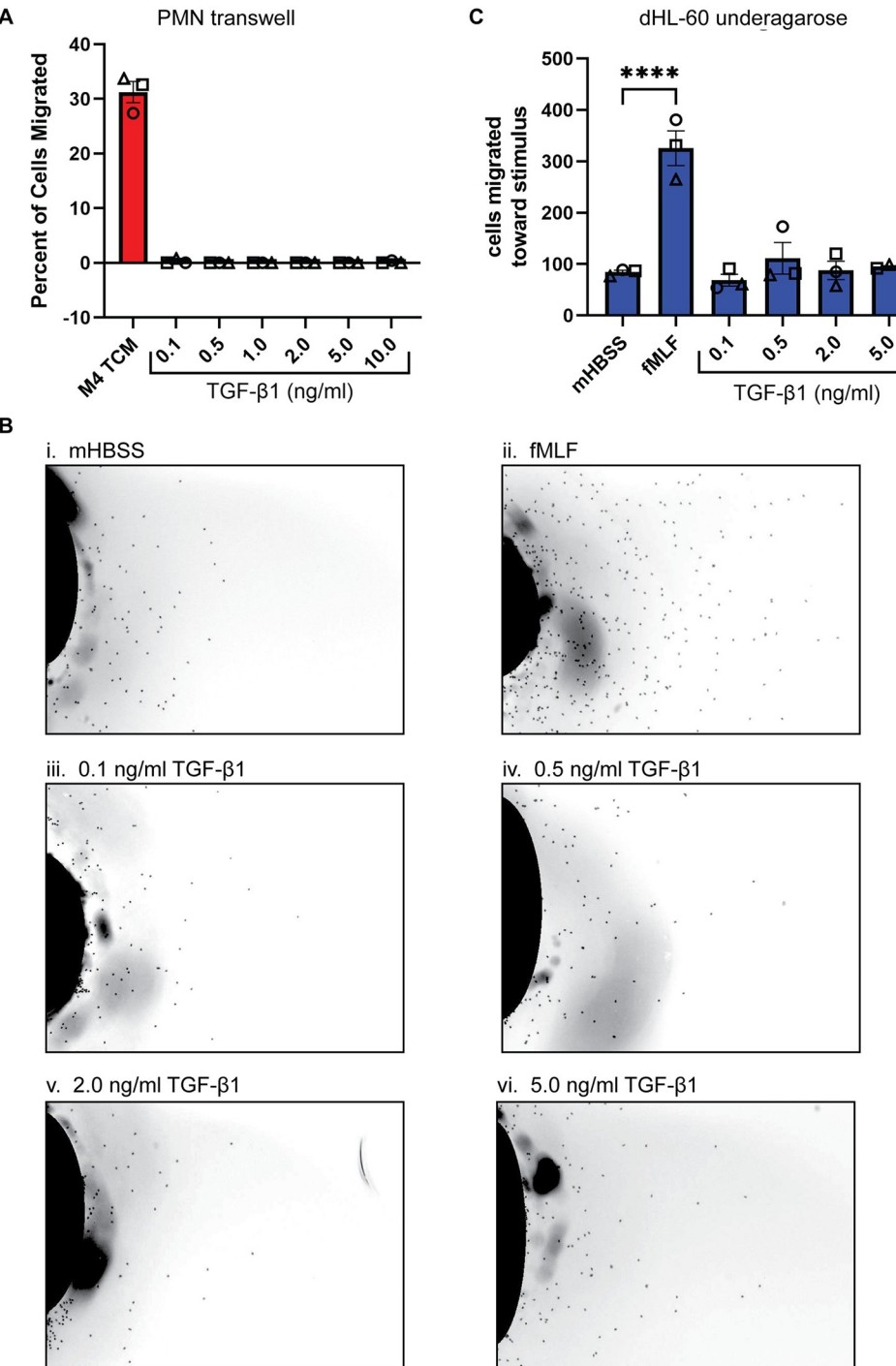

**Fig 1. TGF-β1 does not induce neutrophil migration. (A)** Graph showing the percentage of PMNs that migrated in transwell assays in response to M4 TCM or varying doses of TGF-β1. **(B)** Representative endpoint images of underagarose assays of dHL-60 cells stained with Hoechst. Migration stimuli were (i) mHBSS, (ii) fMLF, (iii-vi) 0.1, 0.5, 2, and 5 ng/ml TGF-β1. **(C)** Quantification of (B) showing the number of PMNs that migrated toward each stimulus. N = 3 independent donors/experiments. Data shown are mean ±SEM with individual dots representing individual donors/experiments (A, C). *P ≤ 0.05 when compared to mHBSS (one-way ANOVA with Dunnett's multiple comparisons test). Raw data are available in S2 File.

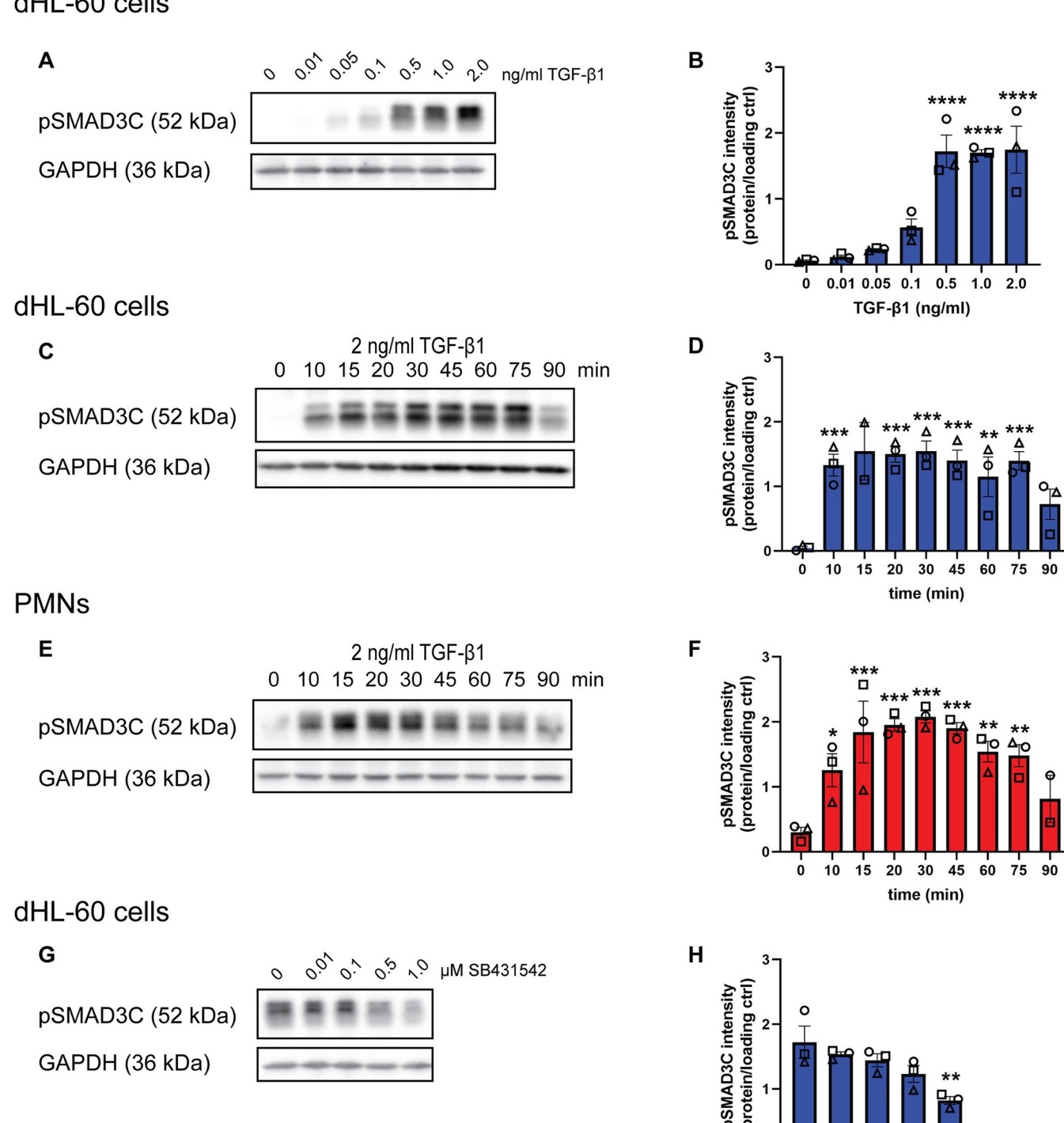

**Fig 2. TGF-β1 induces canonical SMAD3 phosphorylation in neutrophils.** (**A**) Immunoblots for pSMAD3C and GAPDH in dHL-60 cells in response to varying doses of TGF-β1. (**B**) Quantification of (A). (**C**) Immunoblots of pSMAD3C and GAPDH in dHL-60 cells in response to 2 ng/ml TGF-β1 after pretreatment with varying doses of SB431542. (**D**) Quantification of (C). (**E, G**) Immunoblots of pSMAD3C and GAPDH in (E) dHL-60 cells or (G) PMNs in response to 2 ng/ml TGF-β1 from 0 to 90 min. (**F**) Quantification of (E). (**H**) Quantification of (G). All blots are representative of three independent experiments. All quantification graphs depict intensity of the protein of interest normalized to the intensity of the loading control. Data shown are mean ±SEM

with individual dots representing individual experiments. *P ≤ 0.05, **P ≤ 0.01, ***P ≤ 0.0001, ****P ≤ 0.0001 when compared with 0 ng/ml TGF-β1 (B), 0 μM SB431542 (D), or time 0 (F, H) (one-way ANOVA with Dunnett's multiple comparisons test). Raw data are available in S1 and S2 Files.

pSMADC3 and found a dose-dependent inhibition of pSMADC3 levels (Fig 4E and 4F). These findings indicate that the TCM-induced SMAD3 phosphorylation is dependent on TGF-β1. We also measured a significant increase in pERK1/2 in TCM-treated PMNs (Fig 4G and 4H), as reported before [25]. Interestingly, the pERK1/2 levels were sustained over time, unlike the transient response observed with fMLF or the chemokine CXCL1 (S1 Fig) or TGF-β1 alone (Fig 3A and 3B), suggesting a potential synergistic effect between chemokines and TGF-β1 present in the TCM. In contrast, no strong increase in pERK1/2 was detected in TCM-treated dHL-60 cells (Fig 4I and 4J), likely due to the high basal level of pERK1/2 in dHL-60 cells as noted before (Figs 3C and 4D). Together, these findings show that TCM activates SMAD3 and ERK1/2 in PMNs.

### TGF-β1 treatment does not induce the secretion of LTB₄ in neutrophils

Because TCM activates sustained ERK1/2 signaling in neutrophils (Fig 4G and 4H) and ERK1/2 signaling is important for the secretion of leukotriene B₄ (LTB₄) [52, 53], a secondary

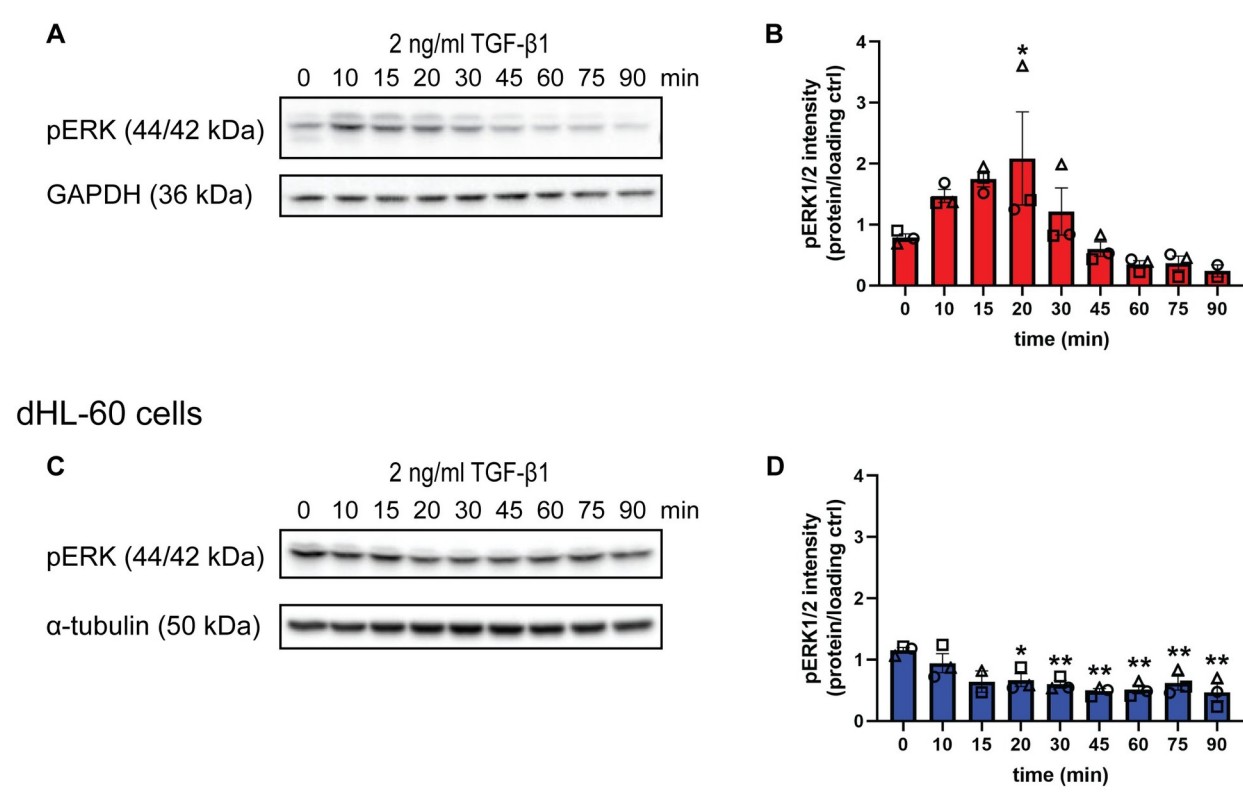

**Fig 3. TGF-β1 noncanonically activates ERK1/2 in primary neutrophils. (A)** Immunoblots of pERK1/2 and GAPDH in PMNs in response to 2 ng/ml TGF-β1 from 0 to 90 min. **(B)** Quantification of (A). **(C)** Immunoblots of pERK1/2 and α-tubulin in dHL-60 cells in response to 2 ng/ml TGF-β1 from 0 to 90 min. **(D)** Quantification of (C). All blots are representative of three independent experiments. All quantification graphs depict intensity of the protein of interest normalized to the intensity of the loading control. Data shown are mean ±SEM with individual dots representing individual experiments. *P ≤ 0.05, **P ≤ 0.01 when compared time 0 (one-way ANOVA with Dunnett's multiple comparisons test). Raw data are available in S1 and S2 Files.

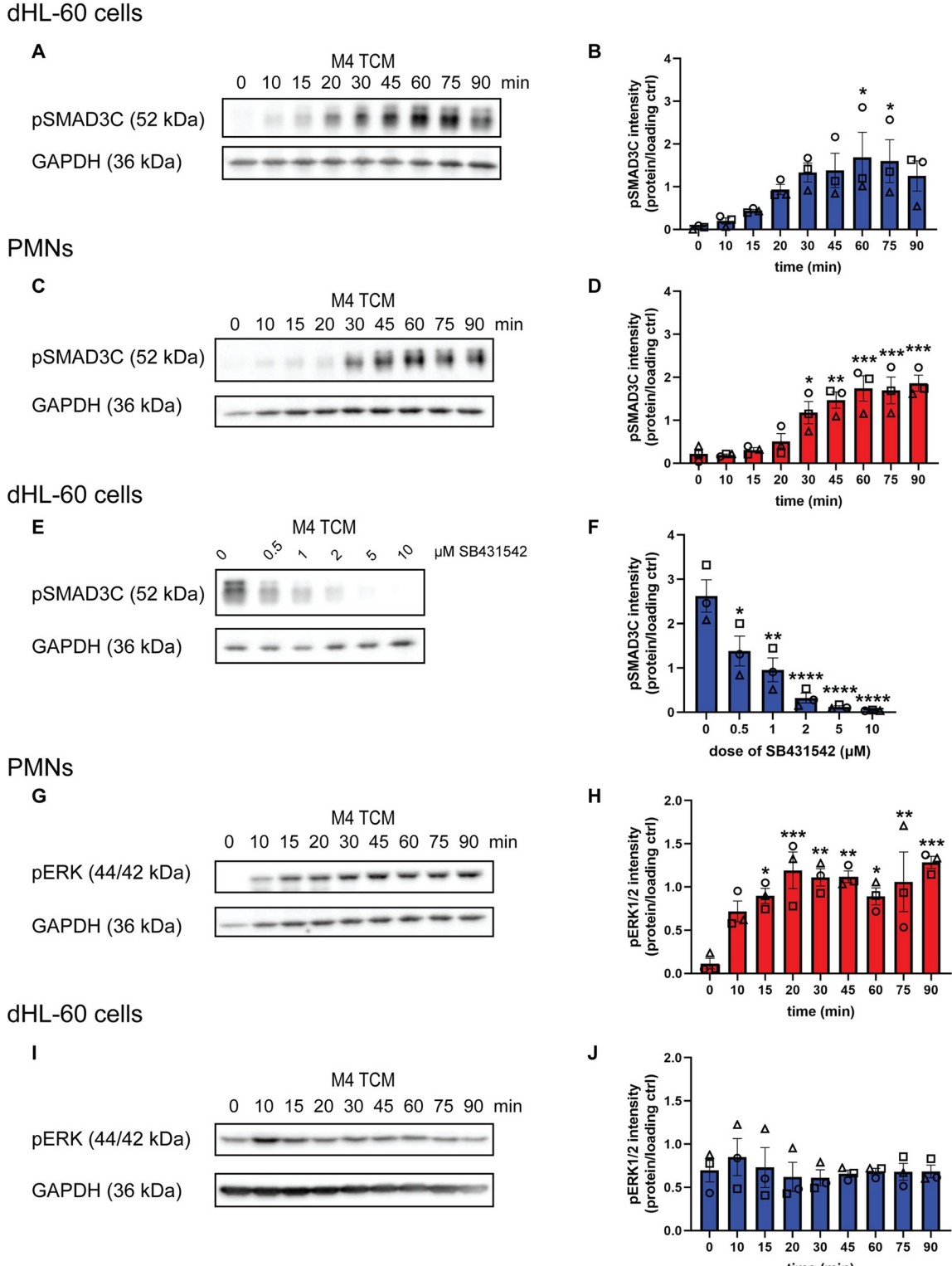

**Fig 4. TGF-β1 present in TCM induces SMAD3 phosphorylation in neutrophils. (A, C)** Immunoblots of pSMAD3C and GAPDH in (A) dHL-60 cells or (C) PMNs in response to M4 TCM from 0 to 90 min. **(B)** Quantification of (A). **(D)** Quantification of (C). **(E)** Immunoblots of pSMAD3C and GAPDH in dHL-60 cells in response to M4 TCM after pretreatment with varying doses of SB431542. **(F)** Quantification of (E). **(G, I)** Immunoblots of pERK1/2 and GAPDH in (G) PMNs or (I) dHL-60 cells in response to M4 TCM from 0 to 90 minutes. **(H)** Quantification of (G). **(J)** Quantification of (I). All blots are representative of three independent experiments. All

quantification graphs depict intensity of the protein of interest normalized to the intensity of the loading control. Data shown are mean ±SEM with individual dots representing individual experiments. *P ≤ 0.05, **P ≤ 0.01, ***P ≤ 0.0001, ****P ≤ 0.0001 when compared with time 0 (one-way ANOVA with Dunnett's multiple comparisons test). Raw data are available in S1 and S2 Files.

chemoattractant critical for neutrophil migration [4], we set out to assess whether TCM induces the secretion LTB$_4$ in PMNs. We treated PMNs with M4 TCM or positive control fMLF for 15 min and measured the LTB$_4$ content in the supernatant. We detected significantly higher levels of LTB$_4$ in the supernatant of PMNs in response to fMLF, compared to the vehicle control DMSO (Fig 5Ai). Interestingly, while LTB$_4$ content in M4 TCM was negligible (average: 9.4 pg/ml, N = 2), neutrophils treated with M4 TCM released significantly higher levels of LTB$_4$ compared to the media control–although LTB$_4$ levels in response to M4 TCM were about ¼ of the levels measured in response to fMLF (Fig 5Aii). To determine which factor(s) in the TCM induces neutrophils to secrete LTB$_4$, we treated PMNs with factors prominently present in the M4 TCM—TGF-β1, CXCL1, or IL-8 [25]. Although PMNs produced on average 750 pg/ml LTB$_4$ in response to the positive control fMLF, no single agent or combination of TGF-β1, CXCL1, or IL-8 stimulated the secretion of LTB$_4$ when compared to the RPMI media control (Fig 5B).

To determine if LTB$_4$ production is important for neutrophils to migrate toward TCM, we pretreated PMNs with MK886 (5-Lipoxygenase Activating Protein (FLAP) inhibitor [35]), SB431542, or AZD5069 (a CXCR2 receptor antagonist [34]) and measured their migration toward M4 TCM using a transwell system. As previously reported, we found that ~40% of PMNs migrate toward M4 TCM in the presence of vehicle control and that this migration was significantly reduced by about half in the presence of SB431542 and AZD5069 (Fig 5C) [25]. PMN migration remained intact in the presence of MK886 alone and when MK886 was paired with SB431542, suggesting that TGF-β signaling and LTB$_4$ production do not coordinate to promote neutrophil migration (Fig 5C). However, similar to the decrease observed when AZD5069 and SB431542 are used together, we observed a decrease in the migration toward TCM when MK886 was used in combination with AZD5069, suggesting that LTB$_4$ production is important for neutrophil migration toward TCM in the absence of chemokine signaling (Fig 5C). When SB431542 is added to MK886 and AZD5069, no further decrease in migration is observed. Taken together, these findings indicate that neutrophils partially rely on LTB$_4$ to migrate toward TCM. However, TGF-β1 by itself or in combination with other factors does not induce neutrophils to secrete LTB$_4$.

## TGF-β1 stimulates the expression of canonical TGF-β targets and pro-tumor genes

We next investigated whether TGF-β1 alters the expression of genes that are relevant for neutrophil migration or tumor-associated functions of neutrophils. We performed bulk RNA-sequencing of dHL-60 cells that were either untreated (time 0) or stimulated for 30 min with TGF-β1, M4 TCM, or their respective media controls IMDM and DEM/F12. The 30 min time-point was selected based on pilot studies assessing the TGF-β1-dependent expression of *PAI1* [54, 55] and *KLF10* [56], two known TGF-β targets (Fig 6A). In addition, to ensure that the cells responded to TGF-β1 before sending the samples for RNA-sequencing, we performed RT-qPCR for *KLF10* and another TGF-β target gene *SMAD7* [57], which showed on average 7.5-fold and 4.3-fold increases, respectively (Fig 6B).

Significant changes in gene expression occurred for all conditions compared to untreated controls. Most remarkably, we found that incubation in plain media (IMDM or DMEM/F12), significantly altered the steady state mRNA levels of 938 and 2879 genes, respectively,

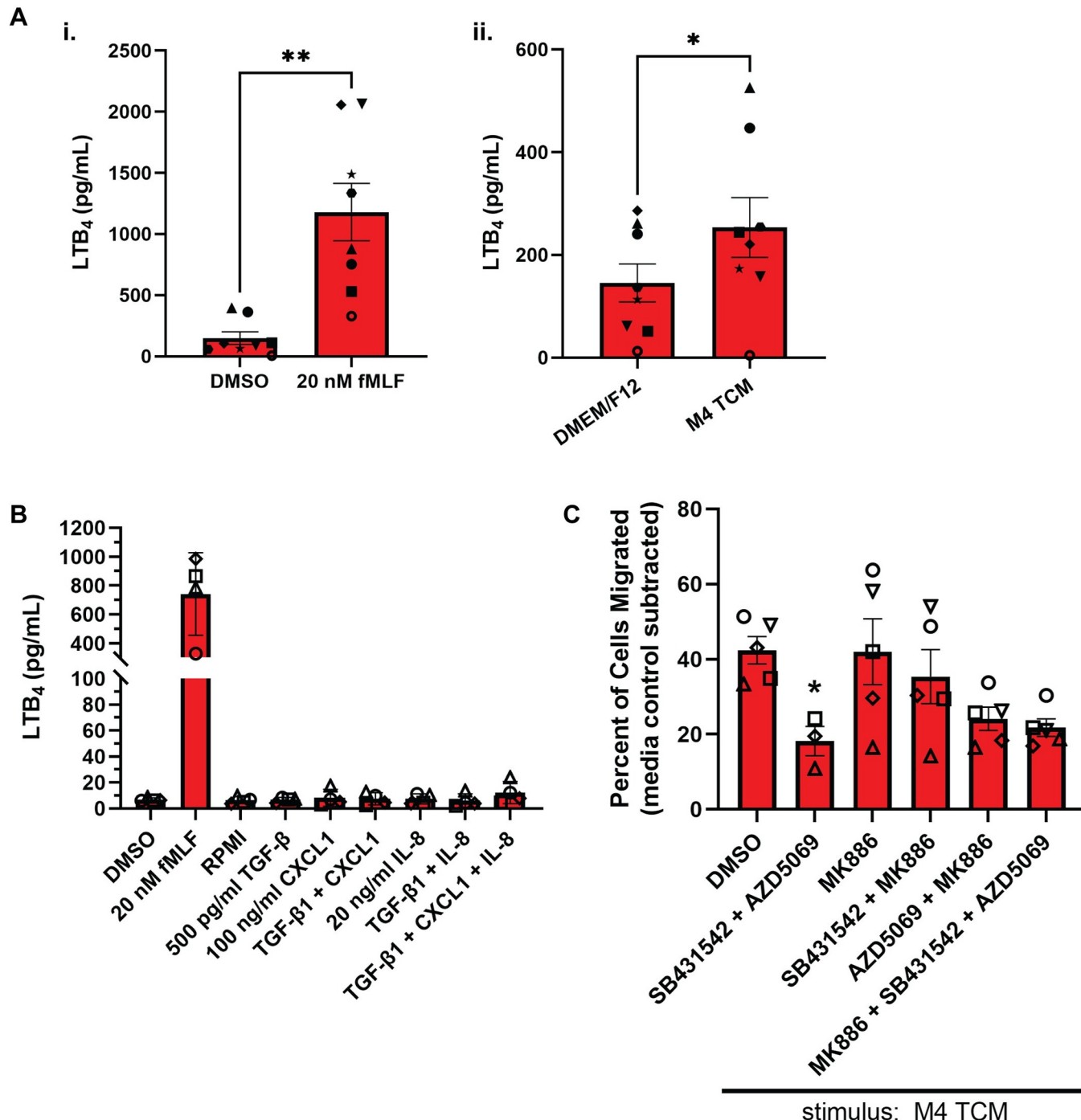

**Fig 5. TGF-β1 does not induce secretion of LTB₄. (A, B)** Graphs depicting secreted LTB₄ in response to various stimuli. N = 8 (Ai, Aii) and N = 4 (B) independent donors. **(B)** 500 pg/ml TGF-β1, 100 ng/ml CXCL1, and 20 ng/ml IL-8 were used whether alone or in combination. **(C)** Graph showing the percentage of PMNs that migrated in transwell assays in response to M4 TCM after pretreatment with various inhibitors. Inhibitors used were 1 μM AZD5069, 500 nM SB431542, and 100 nM MK886. All conditions contain the same amount of vehicle DMSO. N = 3–5 independent donors. Data shown are mean ±SEM with individual dots representing individual experiments. *P ≤ 0.05, **P ≤ 0.01 when compared to DMSO (Ai) or DMEM/F12 (Aii) using paired t-test or when compared to DMSO (C) using one-way ANOVA with Dunnett's multiple comparisons test. Raw data are available in S2 File.

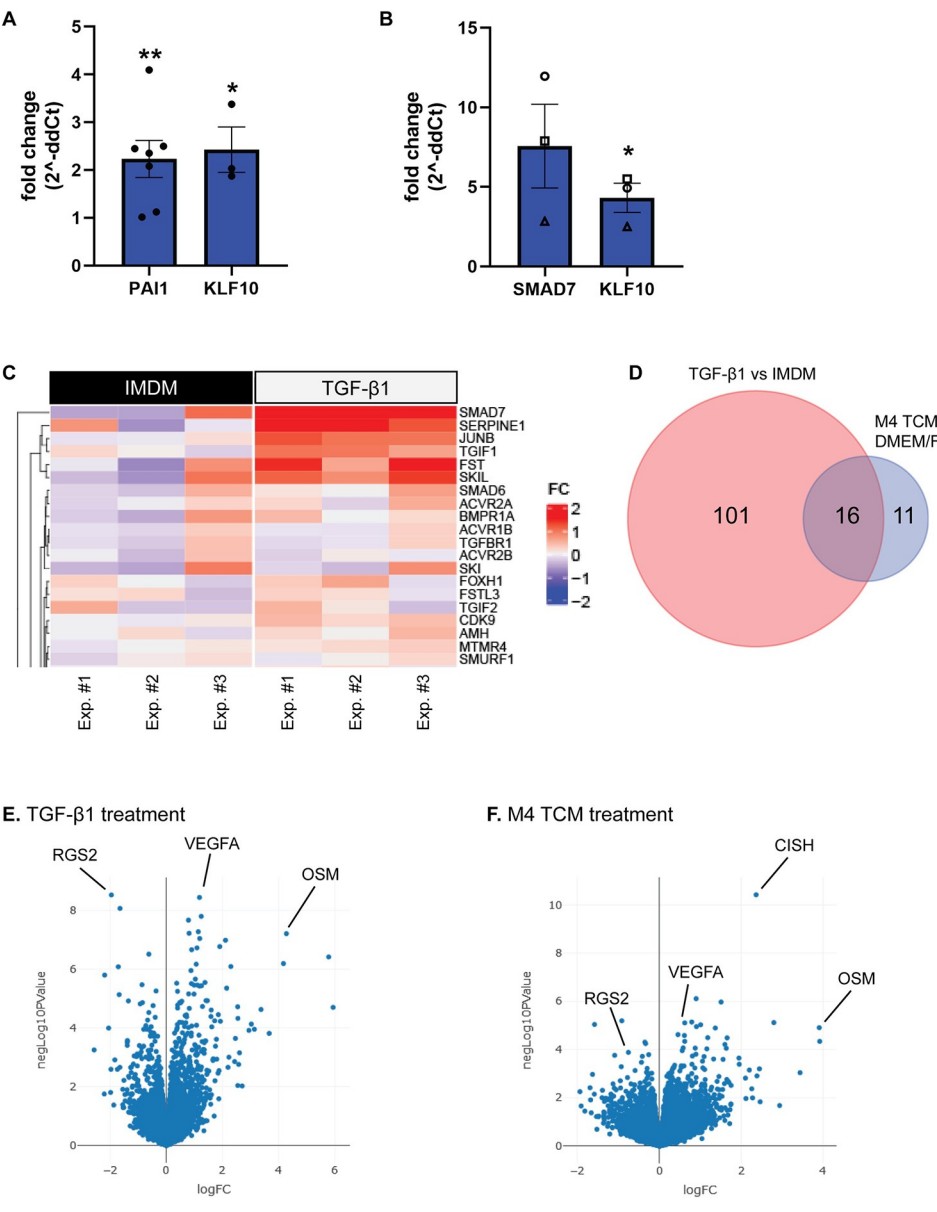

**Fig 6. TGF-β1 and M4 TCM alter gene expression. (A)** RT-qPCR for *PAI1* and *KLF10* expression in dHL-60 cells treated with 2 ng/ml TGF-β1 for 30 min. **(B)** RT-qPCR for *SMAD7* and *KLF10* expression in dHL-60 cells treated with 2 ng/ml TGF-β1 for 30 min that were sent for RNA-seq. Fold changes and statistics are relative to the media control at 30 min (A,B). **(C)** Heat map of Log2 Fold Change to mean IMDM expression for dHL-60 cells treated with either media control (IMDM) or TGF-β1 for 30 min. Plot contains the top 20 average Log2 Fold Change genes from 'signaling by TGF-β family members (R-HSA-9006936)' reactome pathway. **(D)** Venn diagram depicting the number of changed genes in the TGF-β1- and TCM-treated conditions relative to their respective media controls. **(E, F)** Volcano plots of differentially expressed genes identified in the (E) TGF-β1 treatment (IMDM media control) and the (F) M4 TCM treatment (DMEM/F12 media control). *P ≤ 0.05, **P ≤ 0.01 when compared to the time-matched media control using an unpaired t-test (A, B). Raw data are available in S2 File.

suggesting that the composition of the media has effects on gene expression. 829 of these genes overlapped between the two media conditions (S2A Fig). Some of the changes that occurred under all conditions compared to the untreated control clustered in the 'chemokine receptors bind chemokines' and 'neutrophil degranulation' reactome pathways (S2B and S2C Fig).

**Table 3. Genes changed in response to both TGF-β1 and M4 TCM treatment.**

| Gene | TGF-β1 Log2FC | TGF-β1 Adj P-value | M4 TCM Log2FC | M4 TCM Adj P-value |
|---|---|---|---|---|
| **RGS2** | -1.95 | <0.0001 | -0.75 | 0.0794 |
| **VEGFA** | 1.18 | <0.0001 | 0.62 | 0.017 |
| **MIDN** | 1.14 | 0.0001 | 0.62 | 0.0414 |
| **IER2** | 1.25 | 0.0001 | 0.9 | 0.0058 |
| **OSM** | 4.28 | 0.0001 | 3.91 | 0.0174 |
| **KLF10** | 2.11 | 0.0002 | 1.37 | 0.0174 |
| **PLAU** | 0.9 | 0.0003 | 0.61 | 0.0304 |
| **JUNB** | 1.09 | 0.0003 | 0.79 | 0.017 |
| **ID1** | 5.78 | 0.0004 | 3.92 | 0.0414 |
| **DDIT4** | 2.3 | 0.0007 | 1.66 | 0.0334 |
| **TGIF1** | 0.88 | 0.0009 | 0.58 | 0.0603 |
| **CISH** | 1.04 | 0.0016 | 2.36 | <0.0001 |
| **PLK3** | 1.19 | 0.002 | 0.81 | 0.097 |
| **MIR23AHG** | 1.38 | 0.002 | 1.51 | 0.0058 |
| **CEBPB** | 1.11 | 0.002 | 1.01 | 0.017 |
| **ENSG00000260293.2** | 0.89 | 0.0317 | 0.89 | 0.0911 |

In response to TGF-β1 treatment, only 117 genes showed altered expression when compared to its media control IMDM. As expected, analysis of the RNAseq data using the reactome pathway 'signaling by TGF-β family members' revealed that several genes are upregulated in response to TGF-β1 treatment (Fig 6C, S1 Table) and are overrepresented in the list of differentially expressed genes, as confirmed by Fisher's exact test (Table 2). However, no statistical enrichment in pathways involving chemokines and related signaling pathways or in the LTB$_4$ synthesis machinery/arachidonic acid metabolism were detected with the reactome pathway analysis (Table 2). We did observe enrichment in the number of significantly altered transcripts in the 'Class A1 (rhodopsin-like receptors)' reactome pathway, a large subfamily of GPCRs, as well as in the 'signaling by interleukins' reactome pathway (Table 2). The genes that were significantly changed in these pathways are shown in S1 Table.

Only 27 genes showed altered expression in response to M4 TCM treatment relative to its media control DMEM/F12. Remarkably, 16 of these genes were changed in both the TGF-β1 and M4 TCM treatment conditions (Fig 6D, Table 3). Of these 16 genes, the expression of two genes with known roles in the tumor-promoting functions of neutrophils was upregulated: vascular endothelial growth factor A (*VEGFA*) [58–60] and oncostatin M (*OSM*) [18, 61] (Fig 6E and 6F). Furthermore, regulator of G protein signaling 2 (*RGS2*) was downregulated in both treatment conditions, although further investigation is needed to understand the implications of this change in the context of tumor-secreted factors. It is also important to note that cytokine inducible SH2 containing protein (*CISH*) is significantly upregulated in the M4 TCM treatment condition; however, the biological significance in this context is unknown (Fig 6F). Altogether, these results indicate that neutrophil-like cells alter their gene expression in response to TGF-β1 and M4 TCM and upregulate *OSM* and *VEGFA* within 30 min of treatment.

## Discussion

One of several ways TGF-β supports tumor progression is through immunosuppression. For example, TGF-β has been reported to suppress the proliferation and activation of T cells [62] and to skew neutrophils to a tumor-promoting phenotype [20–23]. As most of the studies

investigating the effects of TGF-β on neutrophils in the context of cancer were performed *in vivo*, our goal was to determine the direct effects of TGF-β on neutrophil signaling and migration *in vitro*.

In this study, we used both primary neutrophils and neutrophil-like HL-60 cells to characterize TGF-β signaling pathways, evaluate migration response, and probe for gene expression in response to stimulation with recombinant TGF-β1 or TCM containing TGF-β1. We found that TGF-β1 signals both canonically and noncanonically in neutrophils and activates expression of genes implicated in tumor promotion. However, TGF-β1 alone does not stimulate neutrophil migration nor induce transcription of genes associated with neutrophil migration.

We previously reported a combined effect of tumor-derived TGF-β1 and chemokines in driving neutrophil recruitment induced by conditioned media harvested from aggressive breast cancer cell lines [25]. However, the ability of TGF-β signaling to mediate neutrophil migration remains controversial. For instance, systemic *in vivo* inhibition of TGF-βRI in murine models of lung cancer and mesothelioma results in an influx of neutrophils to the primary tumors [20]. In contrast, in an organoid transplantation model of murine intestinal adenocarcinoma, neutrophil-specific deficiency of TGF-βRI leads to fewer neutrophils in the primary tumors [26]. *In vitro* studies have also led to contradictory findings pertaining to TGF-β1-mediated neutrophil migration. While Reibman *et al.* reported that neutrophils migrated toward TGF-β1 at very low concentrations, with peak migration at 1 pg/ml [28], Shen *et al.* reported that neutrophils do not migrate toward TGF-β1 at concentrations of 1 or 100 pg/ml. Using a wide concentration range of TGF-β1 in both transwell and underagarose migration assays, we now report that neither neutrophils nor dHL-60 cells migrate towards TGF-β1, confirming the findings from Shen *et al.* and showing TGF-β1 signaling does not regulate neutrophil recruitment. We do note that the TGF-β1 concentration range used in our study was higher than Reibman *et al.* While we cannot rule out that lower TGF-β1 concentrations could induce neutrophil migration, we selected the TGF-β1 concentration range based on the amount of TGF-β1 we previously measured in M4 TCM [25], thereby making our findings relevant to our experimental conditions. In addition, we observed that TGF-β1 treatment does not induce changes in the reactome pathways that regulate neutrophil migration/chemotaxis, thereby further validating our functional findings.

LTB$_4$ secretion and relay to neighboring cells is essential to amplify neutrophil recruitment to sites of inflammation and injury [4, 63]. As we found that TCM induces LTB$_4$ secretion in neutrophils, we hypothesized that LTB$_4$ has the potential to regulate neutrophil recruitment to tumors. Indeed, we found that TCM-induced neutrophil chemotaxis is inhibited when LTB$_4$ synthesis is blocked in the presence of a chemokine receptor antagonist. As neutrophils did not secrete LTB$_4$ when stimulated with TGF-β1, CXCL1, or IL-8 –three components present in abundance in M4 TCM–alone or in combination, we reason that the secretion of LTB$_4$ in response to TCM is complex and that further investigation is required to tease apart which factor/s in the TCM are responsible for this response.

TGF-β signals canonically through SMAD2/3 and noncanonically through other pathways, such as PI3K/Akt, MAPKs, and Rho-family GTPases [49]. We found that stimulation of neutrophils with TGF-β1 leads to the phosphorylation of SMAD3, as recently reported by Wang *et al.* [64]. We also discovered that M4 TCM phosphorylates SMAD3 in neutrophils and dHL-60 cells. Although the presence of chemokines such as CCL2 may lead to SMAD3 phosphorylation [65], using a TGF-β1 receptor specific inhibitor, we confirmed that tumor-secreted TGF-β1 is solely responsible for the pSMAD3 response in TCM-treated cells. This SMAD3 phosphorylation is sustained for up to 90 min. Interestingly, we also found that TGF-β1 leads to the phosphorylation of ERK1/2, with a peak around 10–20 min. However, stimulating PMNs with TCM resulted in sustained phosphorylation of ERK1/2, as opposed to a short peak

observed with either TGF-β1 or fMLF. How tumor-secreted TGF-β1 and chemokines coordinate to induce such sustained responses needs to be addressed in future studies because sustained versus transient ERK1/2 phosphorylation could result in different downstream effects. While TGF-β1 and M4 TCM similarly triggered SMAD3 phosphorylation in dHL-60 cells, we detected high basal levels of pERK1/2 in these cells and no further increase in response to these stimuli. This was likely due to the presence of an activating NRAS mutation (Q61L) [51] in HL-60 cells, which has been shown to specifically activate the MEK/ERK pathway in melanocytes [66]. Future studies assessing the ability of TGF-β to signal through other noncanonical signaling pathways in neutrophils are needed. Of particular interest is TGF-β signaling via Rho GTPases, which are key regulators of migration [67].

The RNAseq analysis revealed that extensive alterations in gene expression occurs when dHL-60 cells are incubated in plain media for 30 min. It is unclear if these changes are due to serum-starvation, self-activation, or mild mechanical activation that is likely occurring during rotational incubation. There is also a large difference in the number of genes that show altered expression in response to IMDM media versus DMEM/F12 media, which is likely due to different levels of media components or additional components in the DMEM/F12 media. These changes underscore the importance of using caution while interpreting gene expression alterations in neutrophils in response to various stimuli. After further analysis, we unexpectedly observed no changes in the expression of genes involved in pathways that regulate neutrophils migration. Interestingly, however, the expression of two potentially tumor-promoting genes, *OSM* and *VEGFA*, was upregulated in both the TGF-β1- and M4 TCM-treated dHL-60 cells. Oncostatin M (OSM) is an IL-6 family cytokine that regulates gene expression, inflammation, and cell survival [68, 69]. It has been reported that the expression of *OSM* increases in neutrophils in response to TCM isolated from an aggressive breast cancer cell line and promotes cancer cell detachment and invasion [18]. Other groups have specifically linked GM-CSF stimulation to *OSM* expression in neutrophils [18, 70, 71]. Our data corroborate these findings by showing that M4 TCM, which contains high levels of GM-CSF along with TGF-β1 [25], induces the expression of *OSM*. Moreover, our data show that TGF-β1 alone induces *OSM* expression.

VEGF is important for angiogenesis in cancer [60, 72], and neutrophils have been shown to contribute to angiogenesis in the tumor microenvironment through various mechanisms, including the release of pre-formed VEGF protein [58]. Jablonska *et al.* found that neutrophils isolated from tumor-bearing *Ifnb*[-/-] mice have increased levels of *Vegf* mRNA and contribute to tumor growth and angiogenesis [59]. Additionally, higher intratumoral levels of VEGF in advanced breast cancer is correlated with worse progression-free survival after tamoxifen or chemotherapy treatment [73]. Future studies are needed to specifically determine the function of OSM and VEGF in neutrophils during cancer progression.

In conclusion, this study assessed the ability of TGF-β1 to recruit neutrophils and determined that TGF-β1 by itself is unable to induce neutrophil migration. It is also the first study to conduct kinetics experiments in TGF-β1-stimulated neutrophils to evaluate canonical and non-canonical signaling pathways. Finally, our RNAseq data shed light on how TGF-β1 imparts tumor-promoting functions in neutrophils. Together, this study expands the current understanding of how tumor-secreted factors, specifically TGF-β1, impact neutrophil response and skew neutrophils to support tumor progression, which can inform the design of effective therapeutic strategies against cancer progression and metastasis.

## Supporting information

**S1 Fig. fMLF and CXCL1 stimulate pERK1/2. (A)** Immunoblots of pERK1/2 and GAPDH in dHL-60 cells in response to fMLF or CXCL1 from 0 to 20 min. **(B)** Quantification of (A).

***P ≤ 0.0001, ****P ≤ 0.0001 when compared with time 0 (untreated (UT)) (one-way ANOVA with Dunnett's multiple comparisons test). Raw data are available in S1 and S2 Files. (TIF)

**S2 Fig. Changes in the environment of dHL-60 cells affect gene expression. (A)** Venn diagram depicting the number of changed genes in the IMDM and DMEM/F12 media conditions when each were compared to untreated control (t0). **(B, C)** Heat maps of Log2 Fold Change to mean time 0 expression in the (B) 'chemokine receptors bind chemokines (R-HSA-380108)' reactome pathway and the (C) 'neutrophil degranulation (R-HSA-6798695)' reactome pathway in dHL-60 cells either untreated (time 0) or treated with media controls (IMDM or DMEM/F12), TGF-β1, or M4 TCM for 30 min. Heat maps in (C) depict the 30 pathway genes with the largest average upregulation or the 30 genes with the largest downregulation relative to untreated. (TIF)

**S1 Table. Significantly changed genes within specific reactome pathways.** (DOCX)

**S2 Table. Number of genes significantly changed in each treatment when compared to either the untreated sample or the time-matched, media control.** (DOCX)

**S1 File. PDF containing raw western blots for all figures.** (PDF)

**S2 File. Excel file containing all raw data used to generate plots.** (XLSX)

# Acknowledgments

We thank all the members of the Parent laboratory for their insights and suggestions. We acknowledge Dr. Michael Holinstat and Amanda Prieur from the Platelet Physiology and Pharmacology Core for providing blood draws for this study and thank Peilin Shen for neutrophil isolation and technical assistance.

# Author Contributions

**Conceptualization:** Lauren E. Hein, Shuvasree SenGupta, Carole A. Parent.

**Formal analysis:** Lauren E. Hein, Shuvasree SenGupta, Gaurie Gunasekaran, Craig N. Johnson.

**Funding acquisition:** Carole A. Parent.

**Investigation:** Lauren E. Hein, Shuvasree SenGupta, Gaurie Gunasekaran.

**Methodology:** Lauren E. Hein, Shuvasree SenGupta.

**Resources:** Carole A. Parent.

**Supervision:** Shuvasree SenGupta, Carole A. Parent.

**Validation:** Lauren E. Hein, Shuvasree SenGupta.

**Writing – original draft:** Lauren E. Hein.

**Writing – review & editing:** Lauren E. Hein, Shuvasree SenGupta, Gaurie Gunasekaran, Craig N. Johnson, Carole A. Parent.

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
