## [Decision Letter · Decision Letter 0]

26 Jun 2023

PONE-D-23-16254TGF-β1 activates neutrophil signaling and gene expression but not migrationPLOS ONE

Dear Dr. Parent,

Thank you for submitting your manuscript to PLOS ONE. After careful consideration, we feel that it has merit but does not fully meet PLOS ONE’s publication criteria as it currently stands. Therefore, we invite you to submit a revised version of the manuscript that addresses the points raised during the review process.

We look forward to receiving your revised manuscript.

Kind regards,

Hanna Landenmark

Staff Editor, PLOS ONE

on behalf of 

Ashok Kumar

Journal Requirements:

"This work was supported by funding from the University of Michigan School of Medicine to CAP, NIH grant R01 AI152517 to CAP, and the Rackham Predoctoral Fellowship to LEH."

4. Please expand the acronym “NIH” (as indicated in your financial disclosure) so that it states the name of your funders in full.

7. We note that you have included the phrase “data not shown” in your manuscript. Unfortunately, this does not meet our data sharing requirements. PLOS does not permit references to inaccessible data. We require that authors provide all relevant data within the paper, Supporting Information files, or in an acceptable, public repository. Please add a citation to support this phrase or upload the data that corresponds with these findings to a stable repository (such as Figshare or Dryad) and provide and URLs, DOIs, or accession numbers that may be used to access these data. Or, if the data are not a core part of the research being presented in your study, we ask that you remove the phrase that refers to these data.

8. Please include your full ethics statement in the ‘Methods’ section of your manuscript file. In your statement, please include the full name of the IRB or ethics committee who approved or waived your study, as well as whether or not you obtained informed written or verbal consent. If consent was waived for your study, please include this information in your statement as well. 

Additional Editor Comments:

You manuscript has been reviewed by two experts in the filed. They have suggested some minor changes in the text which can be addressed while submitting the final copy for production. Please ensure these addressed in the final manuscript.

Reviewers' comments:

Reviewer's Responses to Questions

**Comments to the Author**

1. Is the manuscript technically sound, and do the data support the conclusions?

Reviewer #1: Yes

Reviewer #2: Yes

2. Has the statistical analysis been performed appropriately and rigorously? 

Reviewer #1: Yes

Reviewer #2: Yes

3. Have the authors made all data underlying the findings in their manuscript fully available?

Reviewer #1: Yes

Reviewer #2: Yes

4. Is the manuscript presented in an intelligible fashion and written in standard English?

Reviewer #1: Yes

Reviewer #2: Yes

5. Review Comments to the Author

Reviewer #1: To the authors of “TGF-β1 activates neutrophil signaling and gene expression but not migration”

The research work presented aims to explore the role of TGF-β signaling in human neutrophil and neutrophil like cell line HL-60, in-vitro. The experiments are neatly designed and executed. Results interpretation and discussion is precise and easy to understand.

My recommendation: The article can be accepted in current form with few minor corrections mentioned below.

Line 72: CXCR2 has been used as abbreviation without any prior explanation/full form.

Line 299: NRAS mutation; what does NRAS stand for?

Line 315: CXCL1 has no prior explanation.

Fig 1 B V & VI the numbers should be 2.0 and 5.0 ng/ml respectively instead of 2 and 5.

Fig 6a: Legend has no mention of p-values. So, if the fold change is non-significant, how can one base the discussion of a non-significant result?

Protein markers have no unit mentioned in any western blot.

It is great to see the authors acknowledging the effects of different media on the genes’ expression. I wonder if authors would like to make a comment or open a discussion in the scientific community about the effect of media on a system under study? Especially when the effects of media could change the expression of 100s of genes such as in this study.

Reviewer #2: Manuscript by Lauren E. Hein et al describes impact of TGF-β1 on neutrophil signaling, migration, and gene expression. The results convincingly show direct effects of TGF-β on neutrophils signaling and migration in vitro. This manuscript describes a detailed study, the data are clear, the experiments are well done, the paper was very clearly written, figures are comprehensive with error bars, experiments are properly deigned with controls. In sum, this is a high-quality study reporting a fascinating result, which I strongly recommend publishing after addressing minor comments. The authors not only clear the longlisting question whether TGF-β induce neutrophil chemotaxis but also made several significant new findings. The work certainly adds new knowledge to this field, and in my opinion will be of high impact. Thus, I only have a few minor comments.

Line 72: CXCR2? Detail not explained before this.

Line 282: What concentration of TGF-β1 was used for the time course, for both dHL-60 and PMNs, no mention in the figure or the figure legend?

Line 299: What is NRAS, reference 51 and 66 is about it, need some information about it in this paper too.

Figure 2, 3, and 4: What number 52, 36, 44/42, and 50 are? Size of the protein bands in blots in kDa?

Line 315: What is CXCL1? Only in LTB4 ELISA assay experimental section mentioned as a stimuli.

Line325: secretion of LTB4

Line 325 or Figure 5Ai: What concentration of fMLF was used? Is it same like figure 5B, that is 20 nM? PMNs produced on average 750 pg/ml LTB4 in response to the positive control fMLF, in Figure 5Ai it is more than 1000 pg/ml, so higher than 20 nM used?

Line338-39: What concentration of MK886, SB431542, and AZD5069 was used?

Figure 5B: When TGF-β + CXCL1 used, is the concentration same, that is 500 pg/ml for TGF-β and 100 ng/ml for CXCL1? Same for TGF-β + IL8 combination?

Figure 5C: SB is the abbreviation for SB431542, in previous figures full name was used and not the abbreviation, pick one and be consistent.

Line 359: What concentration of TGF-β1 was used to see the fold change expression of PAI1 and KLF10 targets (in Figure 6A)?

Line 366 and Figure S2A: In text it is IMDM and in figure it is MDM.

Figure 6F: What about CISH expression in response to M4 TCM, upregulated or downregulated, pointed out in figure like VEGFA, OSM and RGS2, but no mention in the text.

Line762 and 763: log2 fold or Log2 Fold, pick one and be consistent.

6. PLOS authors have the option to publish the peer review history of their article (what does this mean?). If published, this will include your full peer review and any attached files.

Reviewer #1: No

Reviewer #2: No

---

## [Author Response · Author response to Decision Letter 0]

10 Aug 2023

Please see the response to reviewer document.

---

## [Editor Report · Decision Letter 1]

18 Aug 2023

TGF-β1 activates neutrophil signaling and gene expression but not migration

PONE-D-23-16254R1

Dear Dr. Parent,

We’re pleased to inform you that your manuscript has been judged scientifically suitable for publication and will be formally accepted for publication once it meets all outstanding technical requirements.

Kind regards,

Ashok Kumar, Ph.D.

Academic Editor

PLOS ONE

Additional Editor Comments (optional):

Authors are commended for the through response to reviewers' critiques and revising the manuscript accordingly.
---

## [Editor Report · Acceptance letter]

29 Aug 2023

PONE-D-23-16254R1 

TGF-β1 activates neutrophil signaling and gene expression but not migration 

Dear Dr. Parent:

I'm pleased to inform you that your manuscript has been deemed suitable for publication in PLOS ONE. Congratulations! Your manuscript is now with our production department. 

Kind regards, 

on behalf of

Dr. Ashok Kumar 

Academic Editor

PLOS ONE